# Toward Accurate Respiratory Rate Estimation Using Hybrid Adaptive Filters on Smartwatch PPG

David Pollreisz*, Mostafa Haghi†, and Nima TaheriNejad*†

* TU Wien, Vienna, Austria

† University of Heidelberg, Heidelberg, Germany

E-mail: mostafa.haghi@ziti.uni-heidelberg.de

*Abstract*—The growing adoption of wearable devices has opened new opportunities for unobtrusive health monitoring and medical research. Among various vital signs, Respiratory Rate (RR), often referred to as the forgotten vital sign, is a key parameter, and estimating it using smartwatch-based Photoplethysmogram (PPG) offers a convenient and nonintrusive alternative to conventional techniques. However, motion artifacts significantly impair signal quality, posing a major challenge for accurate RR estimation. In this study, we present a novel algorithm that combines adaptive filtering by employing a Least Mean Squares (LMS) filter guided by a reference signal derived from Empirical Mode Decomposition (EMD), supported by frequency-domain peak detection to robustly estimate RR. Experimental results demonstrate a 71% reduction in mean estimation error and a 56% reduction in standard deviation compared to both adaptive and non-adaptive baseline methods. These improvements show the enhanced accuracy and reliability of the proposed approach. The low estimation error makes this method a practical solution for continuous RR monitoring in everyday settings, enabling broader integration into wearable healthcare applications.

*Index Terms*—photoplethysmography signal, respiratory rate estimation, adaptive filtering, synthetic noise

## I. INTRODUCTION

Wearable devices have become increasingly popular for monitoring physical health [1]–[3], mental well-being [4], [5], and sport [6]–[8]. Despite these advancements, many physiological parameters such as Respiratory Rate (RR) still lack reliable, nonintrusive monitoring solutions suitable for daily-life environments [9], [10]. Direct RR measurement via spirometry or chest belts is often impractical, making indirect approaches using smartwatches highly desirable. These devices commonly include Photoplethysmogram (PPG) sensors, which detect Blood Volume Pulse (BVP) through light absorption changes caused by pulsatile blood flow.

Respiratory activity modulates BVP in subtle ways of introducing low-frequency Baseline Wanderer (BW), altering amplitude via intrathoracic pressure changes, and affecting beat intervals through respiratory sinus arrhythmia [11]. These modulations occur at typical respiratory rates of 12–20 breaths per minute (0.2–0.33 Hz) and are embedded in the PPG signal. Therefore, RR can be estimated using signal processing techniques applied to PPG data. Two primary methods, time-domain and frequency-domain, are often used for this analysis.

The principle of time-domain methods is based on detecting peaks and troughs in the modulated PPG signal [12], often using zero-crossing and threshold-based algorithms to identify respiratory cycles [13]. While straightforward, they are highly sensitive to noise and motion artifacts. In contrast, frequency-domain methods analyze the Inter Beat Interval (IBI) or PPG-derived signals using Fast Fourier Transformation (FFT) or autoregressive modeling [14]. These approaches estimate RR by identifying dominant frequencies within the respiratory range (0.067–1.08 Hz, or 4–65 Beats per Minute (BPM)) and tend to be more robust under real-world conditions [15]. However, under daily life conditions, motion artifacts [16], [17] such as wrist movements during walking or gesturing severely contaminate the signal, often within the same frequency band as respiration [17].

This challenge has driven the development of artifact removal strategies. Unimodal approaches rely solely on PPG signal characteristics, detecting and excluding distorted segments. In these methods, no auxiliary signal is used for the detection and removal of the movement artifact and separating it from the bio-signal of interest. In consequence, they are mainly limited to the detection of the artifacts and cutting those parts of the signal. In the works using this model [10], [16], [18], [19], various properties of the clean signal are used to detect the contaminated part.

In the second approach, the methods are based on the use of acceleration data from embedded sensors to assist in adaptive filtering [20]. Such methods can also filter them and obtain a cleaner signal for further calculations. A particularly effective class of methods used Least Mean Square (LMS) [21], Recursive Least Square (RLS) [22], and Hankel Matrix [23] filtering, where accelerometer signals serve as a reference to isolate motion components. More recent innovations using synthethic reference, such as Empirical Mode Decomposition (EMD) and its variations, offer synthetic reference generation even in the absence of direct acceleration input. These methods decompose a signal into Intrinsic Mode Function (IMF), enabling isolation of motion and respiratory components based on their spectral content. For example, in [24] a second PPG sensor is used and in [25] Complex Empirical Mode Decomposition (CEMD) technique is used to generate a reference signal from the corrupted PPG signal.

Smartwatches offer a clear advantage of unobtrusiveness and socially acceptable use during everyday activities. However,movement artifacts are unavoidable in real-world usage. In this work, we propose a novel approach that combines

frequency-domain RR estimation with an EMD-enhanced adaptive filtering scheme. Unlike conventional methods that directly feed raw accelerometer signals into the LMS filter, we preprocess the acceleration using EMD to improve reference quality and noise separation. We also explore a comprehensive set of configuration variables, including acceleration computation methods, windowing strategies, use of EMD, and the tuning of adaptive filter parameters such as step size and capping thresholds. Through systematic experimentation, we demonstrate a significant improvement in RR estimation reliability, even under extensive motion conditions. Our approach preserves the convenience of smartwatch-based monitoring while addressing one of its most critical technical limitations.

## II. PRINCIPLES OF LMS AND EMD TECHNIQUES

Our proposed method is a novel RR extraction algorithm that combines LMS and EMD, featuring a customized LMS-based motion artifact removal scheme. We briefly review LMS [21] and EMD [25].

### A. LMS Filtering

LMS filtering removes motion artifacts using a three-axis accelerometer as reference input [21]. The raw PPG is first band-pass filtered (0.3–5 Hz) with a 4th-order Butterworth Infinite Impulse Response (IIR) filter. A reference signal is generated via Singular Value Decomposition (SVD) and passed to a modified LMS filter, where coefficients $h(n)$ are updated based on the least mean error $e(n)$. Applying SVD to IMF matrix identifies dominant oscillatory modes. The singular vector corresponding to the largest singular value, representing the mode with the highest energy, is selected as the synthetic reference signal for the adaptive LMS filter. An identical filter is applied in the reference path for adaptive weight adjustment (X-LMS). The core computations of the X-LMS are:

$$y_c(n) = w^T(n) \cdot u(n) \tag{1}$$

$$e(n) = d(n) - y_c(n) \tag{2}$$

$$u_{C*}(n) = \sum_{i=0}^{I-1} c_i^* \cdot u(n - i - M + 1) \tag{3}$$

$$w(n+1) = w(n) + \mu \cdot u_{C*}(n) \cdot e(n) \tag{4}$$

Here, $u(n)$, $y_c(n)$, $e(n)$, and $d(n)$ represent the input, output, error, and desired output, repectively; $\mu$ the step size; $c_i^*$ the compensation coefficients; and $u_{C*}(n)$ the filtered reference. Finally, peak tracking is performed using adaptive thresholds via the Slope Sum Method (SSM).

### B. EMD

If no accelerometer data are available, a reference signal can be derived directly from the corrupted PPG using the CEMD method [25]. The process starts by identifying local extrema in the signal $(x(t) = d(t) = S(n) + N(n))$ and constructing upper ($umax$) and lower ($umin$) envelopes. Their mean, $m(t) = (umax + umin)/2$, is subtracted to obtain

$h(t) = x(t) - m(t)$. Through iterative sifting, $h(t)$ is decomposed into IMFs until it satisfies IMF criteria ($c_1 = h(t)$). The quasi-residue $r(t) = x(t) - c$ is updated until only one extremum remains. We computed the spectrum of each IMF and identified those falling within the typical respiratory frequency band (approximately 0.1–0.4 Hz). IMFs outside this range and within the PPG frequency, often dominated by noise or unrelated physiological activity, were excluded to form the reference noise signal.

For the adaptive step-size LMS algorithm, the step size is updated using the gradient of the error surface to enhance convergence. The filter follows:

$$y(n) = w^T(n)U(n) \tag{5}$$

$$e(n) = d(n) - y(n) \tag{6}$$

$$w(n+1) = w(n) + \mu e(n)u(n) \tag{7}$$

$$\mu(n+1) = \mu(n) + \rho e(n)\gamma^H(n)u(n) \tag{8}$$

$$\gamma(n) = \frac{\mathrm{d}}{\mathrm{d}\mu(n)}(w(n)) \tag{9}$$

Here, $u(n)$ is the input, $y(n)$ the output, $d(n)$ the desired signal, $e(n)$ the error, $w(n)$ the filter coefficients, $\mu$ the step size, $\gamma^H$ the gradient vector, and $\rho$ the learning rate. In our proposed LMS, the weights are calculated using EMD.

## III. PROPOSED METHOD

### A. Baseline Algorithm

The first step of the algorithm is to band-pass filter the PPG signal to remove offset and any noise outside the range of interest. A finite impulse response filter [26] with the order of $N$ is used, where

$$N = \frac{2 * f_s}{25}, \tag{10}$$

in which $f_s$ is the sampling rate and the coefficients of the filter are calculated using [26]

$$b_k = \begin{cases} -1 & for \quad k = 0, ... \frac{N}{2} - 1 \\ 1 & for \quad k = \frac{N}{2}, ..., N - 1 \end{cases} \tag{11}$$

This filter enhances the detection of signal extrema by applying three criteria: (a) the extremum must exceed the mean signal value, (b) it must be at least $0.4f_s$ apart from neighboring extrema, and (c) it must be flanked by two extrema of the opposite type (i.e., a peak between two troughs or vice versa). Once identified, the BW is computed as the average of a peak and its subsequent trough, while Amplitude Modulation (AM) is their amplitude difference. Both features are normalized to the signal's mean. Frequency Modulation (FM) is derived from the interval between consecutive peaks.

To estimate the RR, we propose a new algorithm in which the signal is first detrended, followed by identifying the Dominant Frequency (DF), defined as the dominant frequency peak within the 0.033–2 Hz band. We refer to this frequency-based estimator as Frequency Domain Peak (FDP). We introduce an optional selective fusion technique (Smart Fusion (SFU)) to

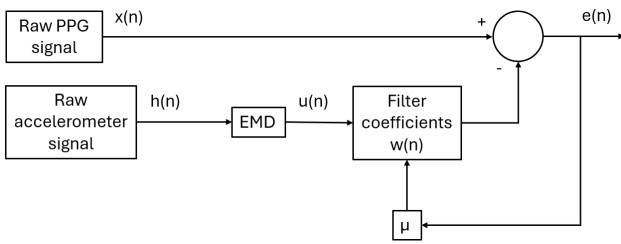

Fig. 1. Flow chart of the proposed adaptive filter.

TABLE I
DISTRIBUTION OF THE RECORDED DATA

|  | No Movement | Movement |
|---|---|---|
| Normal breathing | 10 | 12 |
| Fast breathing | 4 | 4 |
| Slow breathing | 4 | 7 |

Alternatively, the sum of the absolute values of each axis can be computed:

$$\text{Abs}_2 = \sqrt{X^2} + \sqrt{Y^2} + \sqrt{Z^2} \tag{16}$$

This reflects the vector length of acceleration and emphasizes directional components. To further amplify the contribution of larger motions, assuming they induce greater signal corruption, squaring the total acceleration can be used:

$$\text{Acc}^2 = (X + Y + Z)^2 \tag{17}$$

The rationale is that minor accelerations may have a negligible impact, while stronger ones disproportionately affect signal quality. However, in cases where large accelerations might briefly interrupt skin contact, their absolute value becomes less informative. To account for this, we also explore capping the acceleration magnitude to prevent overcompensation in the filtering process. All these modalities are evaluated experimentally to determine the optimal input for noise removal.

## IV. EXPERIMENTAL EVALUATION OF RR EXTRACTION

### A. Dataset

To evaluate the proposed method, 41 signal samples were collected from four male participants aged between 26 and 29 years. All participants provided written informed consent before data collection. The data were recorded using the Empatica E4 smartwatch [27] and included three breathing conditions: normal breathing (10–15 Breaths per Minute (BPM)), fast breathing (>15 BPM), and slow breathing (<10 BPM). During each measurement, the subject raised and lowered their arm from the table to shoulder height and back, repeated three times to induce motion artifacts. A summary of the sample distribution across breathing categories is presented in Table I, and an example of a BVP signal containing the corresponding motion artifacts is shown in Figure 2.

### B. Results of the Proposed Baseline Algorithm

We first evaluated the performance of the baseline algorithm without applying any movement artifact removal (Table II). To determine an optimal window length, we conducted a parameter sweep with window sizes ranging from 4 to 30 seconds, incremented by 2. The upper limit of 30 seconds was chosen because the test recordings are 60 seconds long; exceeding this limit would result in only partial coverage of the signal, as a complete window could no longer be formed. Additionally, the same range of window sizes was tested using 50% overlapping windows.

Comparing RR extraction algorithms is inherently challenging, as performance depends on multiple factors, including

enhance robustness. For each window, the Standard Deviation (STD) of AM, BW, and FM is computed. If the standard deviation of all three features is below 4, their mean is used to estimate the RR. If only two features satisfy this condition and their joint STD is lower than that of all three combined, their mean is used. Otherwise, the SFU returns a *NaN*. We identified this threshold because it offered the best trade-off between excluding noisy windows and retaining enough valid estimates. A brief sensitivity analysis showed that the overall Figure of Merit (FoM) changes by less than ±0.3 across this range, indicating that the method is not highly sensitive to the exact value.

### B. Adaptive Noise Removal

We propose a novel adaptive filtering approach that combines the strengths of both LMS and EMD (Figure 1). While LMS adapts to noise without requiring explicit identification of contaminated segments, EMD enhances performance by generating a reliable reference signal.

The algorithm begins by generating a reference signal from the accelerometer data using EMD. Only components with dominant frequencies within the PPG range (0.1–5 Hz) are retained, as identified by applying the FFT to each IMF.

The adaptive filtering then proceeds according to the following equations:

$$e(n) = x(n) - w(n) \cdot u(n) \tag{12}$$
$$w(n + 1) = w(n) + \mu \cdot e(n) \cdot u(n) \tag{13}$$

Here, $x(n)$ is the corrupted PPG signal, $u(n)$ the EMD-derived reference, $e(n)$ the resulting error and output signal, $w(n)$ the filter weights, and $\mu$ the adaptive step size. The filter is applied across the entire signal. In the absence of motion, $u(n)$ is zero, making $e(n)$ equal to $x(n)$, meaning no filtering is applied. Parameter tuning details are provided in Section IV.

### C. Acceleration Calculation

The acceleration signal used as input to the EMD can be derived from the raw accelerometer data in multiple ways. One basic method is the algebraic sum of the three axes:

$$\text{Norm.} = X + Y + Z \tag{14}$$

An alternative is the magnitude of this sum:

$$\text{Abs.} = \sqrt{(X + Y + Z)^2} \tag{15}$$

TABLE II
STATISTICAL RESULTS OF THE PROPOSED BASELINE ALGORITHM
WITHOUT ANY ADAPTIVE FILTERING.

| Feature | Mean\|Error\| [BPM] | STD | Samples [%] | Window length[s] | Window overlap | FoM |
|---|---|---|---|---|---|---|
| AM | 4.226 | 4.335 | 100 | 16 | | 8.562 |
| BW | 8.157 | 5.009 | 100 | 12 | | 13.168 |
| FM | 4.229 | 3.996 | 100 | 16 | | **8.225** |
| SFU | 3.148 | 3.214 | 75.61 | 28 | | 10.644 |
| AM | 4.568 | 5.524 | 100 | 16 | ✓ | 10.092 |
| BW | 7.838 | 5.261 | 100 | 16 | ✓ | 13.099 |
| FM | 4.567 | 4.253 | 100 | 20 | ✓ | 8.821 |
| SFU | 3.351 | 3.387 | 75.61 | 28 | ✓ | 11.022 |

estimation error, STD, and successfully estimated samples. Therefore, direct comparison between different algorithms is not straightforward. Relying on a single metric may not capture the overall effectiveness of an algorithm. To address this, we used the Figure of Merit (FoM) that integrates these key aspects into a single, interpretable score. The FoM is designed to provide a balanced view of accuracy, robustness, and reliability, giving a more comprehensive and fair comparison across methods. Lower FoM values indicate better performance. That is;

$$FoM = Mean(|Error|) + STD + 10 \times (1 - CSR^2) \quad (18)$$

where $CSR$ denotes the Computed Samples Ratio, defined as the number of successfully estimated RR samples divided by the total number of samples; lower FoM values indicate better algorithm performance.

As shown in Table II, the best-performing algorithm is FM without overlapping windows, achieving the lowest FoM of 8.225. Closely following is AM, also without overlapping windows, with a FoM of 8.562. For both algorithms, the optimal window length is 16 seconds.

### C. Results of the Proposed Adaptive Algorithm

For the adaptive algorithm, we evaluated multiple combinations (Table III). To get the optimal $\mu$, window length, and acceleration cap value, we performed a parameter sweep. In Table IV, the three parameters and their sweeping ranges are shown, where $\mu$ is the adaptive constant. In Table V the best feature of each combination is displayed.

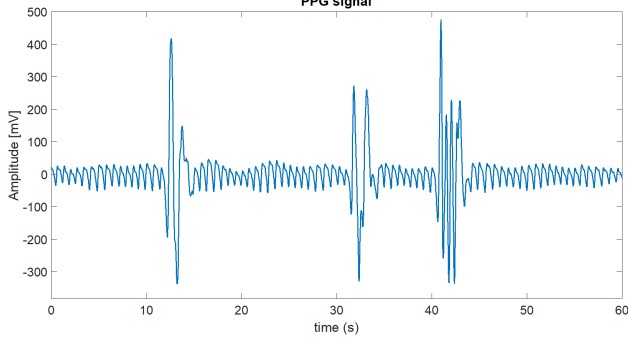

Fig. 2. A BVP signal with three movement artefacts.

TABLE III
A SUMMARY OF ALL TESTED COMBINATIONS FOR THE PROPOSED
METHOD.

| | Acceleration | | | | EMD | Acc. cap | Window overlap |
|---|---|---|---|---|---|---|---|
| | Norm. | Abs. | Acc$^2$ | Abs$_2$ | | | |
| Comb 1 | ✓ | | | | ✓ | | |
| Comb 2 | ✓ | | | | ✓ | ✓ | |
| Comb 3 | | ✓ | | | | | |
| Comb 4 | | ✓ | | | | ✓ | |
| Comb 5 | | ✓ | | | ✓ | ✓ | |
| Comb 6 | | | ✓ | | ✓ | ✓ | |
| Comb 7 | ✓ | | | | ✓ | | ✓ |
| Comb 8 | ✓ | | | | ✓ | ✓ | ✓ |
| Comb 9 | | ✓ | | | | | ✓ |
| Comb 10 | | ✓ | | | | ✓ | ✓ |
| Comb 11 | | ✓ | | | ✓ | ✓ | ✓ |
| Comb 12 | | | ✓ | | ✓ | ✓ | ✓ |
| Comb 13 | | | | ✓ | ✓ | ✓ | ✓ |

The best performance was achieved by Comb 13, which employs "$Abs_2$" (Equation 16) for acceleration computation, incorporates both EMD and an acceleration cap, and uses 50% overlapping windows, resulting in a FoM of 5.663. Figure 3 illustrates a 4D plot where color intensity represents the FoM; darker colors indicate lower FoM values and thus better performance. Two performance peaks are one around a window length of 16, $\mu = 3$, and an acceleration cap of 0.2; and another at a window length of 22, $\mu = 1.5$, and an acceleration cap of 0.7, the latter yielding overall better performance. It should be noted that each method was evaluated over a full parameter sweep derived from the ranges in Table IV. Table V reports the best-performing configuration for each method, selected from this grid.

### V. COMPARISON

#### A. Non-adaptive Algorithms

To have a fair comparison, we implemented three of the most relevant RR extraction algorithms and ran them on our dataset. The first, Time Domain Peak Detection (TDPD) [28], relies on the Peak Detection (PD) method for RR estimation and uses SFU for signal fusion. The second algorithm, Time Domain - Count Origin (TDCO) [12], also employs SFU for fusion but detects peaks and troughs using the Count Origin (CO) method. A threshold is defined as 0.2 times the 75th percentile of the peak amplitudes; any peaks below this threshold are discarded. A breath is identified when two consecutive peaks are separated by a single trough whose amplitude is below zero. The third algorithm, Count Origin - Smart and Time Fusion (COSTF) [12], builds upon the

TABLE IV
SWEEP RANGE OF DIFFERENT PARAMETERS.

| | Start | End | Step size |
|---|---|---|---|
| $\mu$ | 0.1 | 4.1 | 0.1 |
| Acceleration cap | 0.1 | 0.85 | 0.05 |
| Window length | 4 | 30 | 2 |

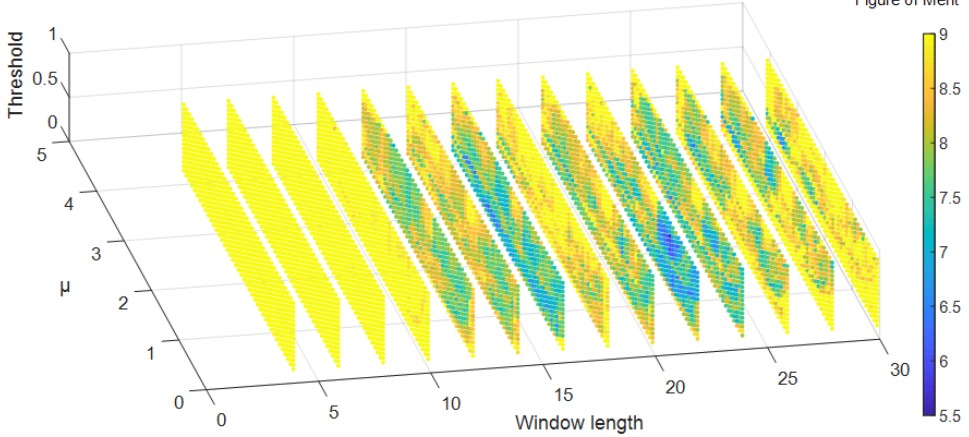

Fig. 3. A 4D plot of the best combination (#13), where the color represents the FoM.

TABLE V
THE BEST RESULTS OF THE PROPOSED ADAPTIVE ALGORITHM FOR EACH
COMBINATION SHOWN IN TABLE III.

| | Feat--ure | Mean\|Err.\| [BPM] | STD | Samples [%] | $\mu$ | Acc. cap | Window length[s] | FoM |
|---|---|---|---|---|---|---|---|---|
| Comb1 | AM | 3.986 | 3.130 | 100 | 0.3 | NA | 12 | 7.116 |
| Comb2 | AM | 3.428 | 2.573 | 100 | 0.3 | 0.55 | 12 | 6.002 |
| Comb3 | FM | 3.321 | 3.450 | 100 | 1.7 | NA | 16 | 6.772 |
| Comb4 | FM | 3.331 | 2.541 | 100 | 0.5 | 0.15 | 16 | 5.872 |
| Comb5 | FM | 3.298 | 2.614 | 100 | 0.8 | 0.15 | 16 | 5.910 |
| Comb6 | AM | 3.708 | 2.839 | 100 | 2.4 | 0.2 | 12 | 6.457 |
| Comb7 | FM | 4.094 | 3.384 | 100 | 0.8 | NA | 16 | 7.478 |
| Comb8 | FM | 3.889 | 2.985 | 100 | 3.0 | 0.7 | 22 | 6.874 |
| Comb9 | FM | 3.936 | 3.206 | 100 | 0.8 | NA | 16 | 7.143 |
| Comb10 | FM | 3.237 | 2.999 | 100 | 1.6 | 0.1 | 24 | 6.235 |
| Comb11 | AM | 3.014 | 2.737 | 100 | 1.4 | 0.2 | 24 | 5.751 |
| Comb12 | AM | 3.696 | 3.147 | 100 | 0.6 | 0.3 | 24 | 6.843 |
| Comb13 | FM | 3.095 | 2.590 | 100 | 1.1 | 0.7 | 22 | **5.663** |

TABLE VI
STATISTICAL RESULTS OF THE RE-IMPLEMENTED TDPD ALGORITHM ON
OUR DATASET WITHOUT ADAPTIVE FILTERING [28].

| | Mean\|Error\| [BPM] | STD | Samples [%] | Window length [s] | Window Overlap | FoM |
|---|---|---|---|---|---|---|
| AM | 15.923 | 9.032 | 9.76 | 8 | | 25.498 |
| BW | 12.248 | 0 | 2.44 | 8 | | 22.242 |
| FM | 10.738 | 6.921 | 92.7 | 24 | | 19.068 |
| SFU | 9.226 | 0 | 2.44 | 12 | | 19.220 |
| AM | 16.222 | 10.001 | 9.76 | 8 | ✓ | 27.182 |
| BW | 17.590 | 6.915 | 2.44 | 8 | ✓ | 25.457 |
| FM | 10.656 | 6.911 | 92.7 | 24 | ✓ | 18.977 |
| SFU | 6.877 | 0 | 2.44 | 12 | ✓ | **16.871** |

TABLE VII
THE BEST STATISTICAL RESULTS OF THE RE-IMPLEMENTED TDCO (TOP
VIEW) [12] AND COSTF (BOTTOM VIEW) [12] ALGORITHMS ON OUR
DATASET WITHOUT ADAPTIVE FILTERING.

| | Mean\|Error\| [BPM] | STD | Samples [%] | Window length [s] | Window Overlap | FoM |
|---|---|---|---|---|---|---|
| AM | 14.618 | 6.914 | 100 | 28 | | 21.562 |
| BW | 16.789 | 6.057 | 100 | 30 | | 22.846 |
| FM | 13.937 | 6.450 | 100 | 24 | | **20.387** |
| SFU | 15.418 | 6.353 | 97.6 | 22 | | 22.253 |
| TFU | 16.257 | 6.753 | 100 | 28 | | 23.009 |
| AM | 14.589 | 5.883 | 100 | 28 | ✓ | 21.322 |
| BW | 17.154 | 5.883 | 100 | 22 | ✓ | 23.037 |
| FM | 13.924 | 6.523 | 100 | 22 | ✓ | 20.447 |
| SFU | 15.323 | 6.156 | 95.1 | 22 | ✓ | 22.431 |
| TFU | 15.676 | 6.877 | 100 | 30 | ✓ | 22.552 |

CO method and incorporates both SFU and Temporal Fusion (TFU). The temporal fusion unit (TFU) functions similarly to a low-pass filter and is computed as follows:

$$RR_i = 0.2RR_{est} + 0.8RR_{i-1} \qquad (19)$$

This formulation helps reduce errors, particularly in the presence of outliers [29].

The results of these algorithms applied to our dataset are presented in Table VI and Table VII. Table VI includes a few instances where the STD is zero. This occurs when only a single RR sample was estimated, making STD undefined. To calculate the FoM, a STD of zero was assigned in these cases. Overall, TDPD demonstrated a very low rate of successful RR estimation, indicating its unreliability. In contrast, TDCO and COSTF achieved nearly 100% success across all test cases. However, both exhibited slightly higher STDs than the proposed method and significantly larger mean errors. When evaluating the FoM, the proposed method outperformed all other algorithms by a margin of approximately 58–70%. TDCO and COSTF, which rely on amplitude and base-line morphology, showed reduced performance after LMS + EMD filtering, likely because the filter alters low-frequency components essential for peak detection, smoothing envelope features, and impairing cycle counting.

### B. Adaptive Algorithms

The majority of adaptive filtering methods aim to remove motion artifacts from the PPG signal to estimate heart rate, rather than RR. For a fair comparison, we applied our adaptive motion artifact removal scheme to the three RR extraction algorithms discussed above. Table VIII and Table IX summa-

TABLE VIII
THE BEST PERFORMANCE OF THE RE-IMPLEMENTED TDPD ALGORITHM
ON OUR DATASET ACROSS ALL PARAMETER COMBINATIONS WITH
ADAPTIVE FILTERING.

| | Feat-ure | Mean\|Err.\|[BPM] | STD | Samples[%] | $\mu$ | Acc. cap | Window length[s] | FoM |
|---|---|---|---|---|---|---|---|---|
| Comb1 | FM | 2 | 0 | 2.44 | 3.6 | NA | 4 | 11.994 |
| Comb2 | SFU | 1.461 | 0 | 2.44 | 2.6 | 0.6 | 12 | 11.456 |
| Comb3 | SFU | 0.408 | 0 | 2.44 | 1.3 | NA | 12 | 10.403 |
| Comb4 | SFU | 0.034 | 0 | 2.44 | 1.2 | 0.8 | 12 | 10.027 |
| Comb5 | FM | 2.945 | 0 | 2.44 | 2.9 | 0.15 | 8 | 12.939 |
| Comb6 | FM | 1.603 | 0 | 2.44 | 0.8 | 0.85 | 16 | 11.597 |
| Comb7 | SFU | 0.002 | 0 | 2.44 | 0.2 | NA | 14 | **9.996** |
| Comb8 | SFU | 0.002 | 0 | 2.44 | 0.2 | 0.1 | 14 | **9.996** |
| Comb9 | SFU | 0.002 | 0 | 2.44 | 0.2 | 0.2 | 14 | **9.996** |
| Comb10 | SFU | 0.002 | 0 | 2.44 | 0.1 | 0.7 | 14 | **9.996** |
| Comb11 | SFU | 0.002 | 0 | 2.44 | 0.1 | 0.1 | 14 | **9.996** |
| Comb12 | SFU | 0.110 | 0 | 2.44 | 0.1 | 0.1 | 14 | 10.109 |
| Comb13 | SFU | 0.002 | 0 | 2.44 | 0.1 | 0.1 | 14 | **9.996** |

TABLE IX
THE BEST RESULTS OF TDCO AND COSTF USING ADAPTIVE FILTERING.
NOTE THAT FOR EACH "COMB" WE HAVE ONLY PRESENTED THE BEST
RESULT OF EITHER OF THE ALGORITHMS. SPECIFICALLY, THE
BEST-PERFORMING METHODS PER COMBINATION WERE: TDCO FOR
COMB1, 3, 6–8, 10, AND 13; AND COSTF FOR COMB2, 4–5, 9, AND
11–12.

| | Feat-ure | Mean\|Err.\|[BPM] | STD | Samples[%] | $\mu$ | Acc. cap | Window length[s] | FoM |
|---|---|---|---|---|---|---|---|---|
| Comb1 | SFU | 1.846 | 0 | 2.44 | 3.8 | NA | 6 | **11.840** |
| Comb2 | SFU | 1.846 | 0 | 2.44 | 3.8 | 0.1 | 6 | **11.840** |
| Comb3 | SFU | 4.634 | 0 | 2.44 | 0.3 | NA | 6 | 14.630 |
| Comb4 | SFU | 4.634 | 0 | 2.44 | 0.1 | 0.25 | 6 | 14.630 |
| Comb5 | SFU | 3.783 | 0 | 2.44 | 3.6 | 0.2 | 6 | 13.778 |
| Comb6 | SFU | 3.473 | 0 | 2.44 | 0.2 | 0.15 | 6 | 13.467 |
| Comb7 | SFU | 4.841 | 0 | 2.44 | 3.7 | NA | 8 | 14.834 |
| Comb8 | SFU | 4.875 | 0 | 2.44 | 3.8 | 0.2 | 8 | 14.860 |
| Comb9 | AM | 13.084 | 5.874 | 100 | 0.2 | NA | 22 | 18.958 |
| Comb10 | FM | 5.095 | 0 | 2.44 | 0.1 | 0.1 | 6 | 15.089 |
| Comb11 | FM | 5.095 | 0 | 2.44 | 0.1 | 0.35 | 6 | 15.089 |
| Comb12 | FM | 5.095 | 0 | 2.44 | 0.6 | 0.15 | 6 | 15.089 |
| Comb13 | FM | 4.185 | 0 | 2.44 | 0.5 | 0.2 | 6 | 14.179 |

rize the best performance of these algorithms under the same conditions.

As shown in the results, the proposed algorithm continues to outperform all other methods by a substantial margin, demonstrating over 43% improvement in FoM, and up to 70% in some cases. Interestingly, the application of adaptive motion artifact removal appears to have a predominantly negative impact on the performance of the other algorithms, which were originally not designed with adaptive filtering in mind. Although adaptive filtering effectively reduced the mean error, it severely compromised the number of successful RR estimations. Except for Comb 9 in both TDCO and COSTF, all other configurations resulted in only a single successful estimation. In this specific combination, performance improved across all metrics: mean error decreased, STD was slightly reduced, and the FoM showed an 11% improvement compared to the same algorithm without adaptive filtering.

### C. Adaptive vs. Non-Adaptive

As discussed above, both proposed algorithms of adaptive and non-adaptive, outperformed their respective counterparts. While the adaptive version incurs additional computational cost due to motion artifact removal, this cost is justified by improved reliability and performance. Specifically, the FoM improved from 8.225 in the baseline algorithm to 5.663 with the adaptive version, corresponding to a 27% reduction in mean error and a 35% reduction in STD. Depending on resource availability, one can choose between the adaptive version for higher accuracy or the base version for efficiency. The proposed non-adaptive algorithm still outperformed the adaptive versions of existing algorithms. Its FoM is at least 18% lower than any of their adaptive counterparts. Given that most of these alternatives exhibit an unacceptably low rate of successful sample estimation, a fair comparison can only be made with Comb 9 of TDCO and COSTF. In that case, our algorithm achieved a 57% lower FoM, 67% lower mean error, and 32% lower STD. These results indicated the strength of the proposed method. The base version stood out as the preferred choice among both adaptive and non-adaptive algorithms, surpassed only by its adaptive counterpart, which came at the cost of increased computation.

Even though tested on a small dataset, our framework was designed with longitudinal, real-world monitoring in mind. We conducted a requirement analysis emphasizing adaptability to signal variability and resilience to motion artefacts. Specifically, the adaptive LMS filter guided by an EMD-derived respiratory reference was introduced to continuously suppress motion artefacts without requiring retraining, while SFU selectively suppresses unreliable outputs based on consistency checks across amplitude, baseline, and frequency dynamics. These components aim to ensure robustness even when the respiratory signal quality fluctuates over time. Our initial short-term validation suggests that the algorithm maintains stable performance under moderate variability. However, continuous daily-life data will introduce more severe and sustained challenges that might require refining the algorithm under these more demanding conditions.

This study was conducted on a small, homogeneous group of healthy male participants under regular breathing conditions to enable initial validation. It included motion conditions specifically, controlled arm-raising and lowering movements, to isolate the effects of motion artifacts in a repeatable manner and still introducing realistic disruptions to the PPG signal. While this facilitated reproducibility, it limits the generalizability of our findings. Broader validation on more diverse cohorts and irregular respiratory patterns (e.g., apnea, dyspnea) is needed to assess algorithmic robustness, which we aim to pursue in future work. The larger dataset enables us to benchmark our approach against lightweight deep learning models, such as ResNet-based RR estimators, to evaluate generalizability and compare performance with data-driven methods.

### VI. CONCLUSION

In this work, we introduced an adaptive algorithm for respiratory rate extraction from smartwatch-acquired PPG signals. The proposed approach employed an adaptive LMS filter, guided by a reference signal generated via EMD, to remove motion artifacts. Unlike many prior studies that rely on time-domain methods, we used a frequency-domain strategy

(DF) for RR estimation, which proved effective even without artifact removal. The base version of our algorithm, without adaptive filtering, already outperformed existing methods that include artifact removal. Incorporating adaptive filtering further enhanced its performance, yielding the best results among all tested algorithms. Overall, the proposed adaptive method achieves up to 70% improvement in FoM compared to existing approaches, with a mean error of just 3.095 (71% lower) and a standard deviation of 2.590 (56% lower), demonstrating both high accuracy and reliability.

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
