# OpenReview forum: "Toward Accurate Respiratory Rate Estimation Using Hybrid Adaptive Filters on Smartwatch PPG"
_IEEE.org/EMBS/BHI/2025/Conference — BHI 2025_

### Official Review · Reviewer_kQFa · 2025-07-17
**Hybrid LMS + EMD Filtering for Smartwatch RR Estimation, but Validation is Limited by Dataset Size and Baseline Scope**

**Confidence:** 3
**Clarity Of Writing:** good
**Clinical Significance:** fair
**Methodological Novelty:** fair
**Overall Rating:** 5
**Final Rating:** 6

**Experiments And Results:**

fair

**Questions For The Authors:**

- Could you scale your dataset to ≥10 subjects with varied demographics?

- Can you compare your method to a lightweight deep model (e.g., the ResNet-based RR estimator from Bian et al., EMBC 2020) trained on your dataset?

**Strengths:**

- Large error/SD reduction vs. baseline: 71% and 56% lower, respectively (Table V)

- Hybrid LMS+EMD structure (Fig. 1, Eq. 12–13) is well-justified and presents a conceptually interesting design

**Summary Of The Paper:**

Proposes a smartwatch RR estimator using FDP and adaptive LMS+EMD filtering. On 41 recordings, adaptive filtering yields 71% lower error vs. baseline. Comparisons and statistical quantification are limited.

**Weaknesses:**

- Very small dataset (4 subjects, 41 mins) with no age/sex/pathology variation (Table I)

- Lacks modern ML baselines (e.g., Deep PPG, CNN-based models), evaluation is limited to classical signal-processing methods

- No confidence intervals or statistical tests, reported gains lack significance validation

---

### Official Review · Reviewer_niK1 · 2025-07-17
**promising method to improve signal quality in smartwatches and fitness trackers, but only small study**

**Confidence:** 3
**Clarity Of Writing:** great
**Clinical Significance:** fair
**Methodological Novelty:** good
**Overall Rating:** 4
**Final Rating:** 5

**Experiments And Results:**

fair

**Questions For The Authors:**

- Please highlight (e.g. bold font) the best results in the result tables (II, VI, VII).
- Did you have a reference sensor for the breathing rate (chest belt?) or how did you ensure, that the subjects kept the instructed breathing rate?

**Strengths:**

Because of the wide popularity of smartwatches or fitness trackers the possiblity to use their data suggests itself.
To improve the quality of the source signal and also the significance of this data to health assessment it should be reliable.
The presented paper focuses on this issue.

- Detailed explanaintion of used methods and algorithm
- Assessment of several competing methods

**Summary Of The Paper:**

The paper introduces a method to reduce the impact of (motion) artefacts in PPG signals of smartwatches and wristbands.
The method makes use of adaptive filtering and an adaptive reference function.
Itsignificantly reduces the effect of artefacts in PPG on respiration rate estimation.
The method works with or without accelerometer data, whereas when this additional data is present, the achieved quality improvement is stronger.

**Weaknesses:**

- Dataset is very small (4 subjects, all male, all young and presumably heathy)
- Study protocol did not feature everyday activities, which introduce motion artefacts (e.g. walking, climbing stairs, cycling, typing, ...)
- Result tables are hard to grasp, maybe you could condense the information more

---

### Official Review · Reviewer_857s · 2025-07-17
**Interesting work with extensive experimentation**

**Confidence:** 4
**Clarity Of Writing:** fair
**Clinical Significance:** good
**Methodological Novelty:** good
**Overall Rating:** 5

**Experiments And Results:**

fair

**Questions For The Authors:**

n/a

**Strengths:**

- Robust respiratory rate signal extraction
- Extensive experimentation
- Comparison to the state of the art

**Summary Of The Paper:**

This paper presents a novel method for estimating respiratory rate (RR) from smartwatch-based photoplethysmogram (PPG) signals. The authors address the significant challenge of motion artifacts, which often corrupt PPG signals in real-world settings.

**Weaknesses:**

- Section III.A: It is unclear how the threshold for the standard deviation was decided. How sensitive is the method to this threshold?
- The use of data from only 4 participants significantly limits the ability to draw safe conclusions from the study. Why didn't the authors collect more data?
- I have some concerns regarding the creation of the ground truth data. I assume that the Empatica smartwatch uses PPG in order to decide the breathing condition of the user. If that is the case, how reliable can the ground truth labels be considered? The presented findings would only be as reliable as the Empatica's RR detection algorithm. In addition, any result apart from an error of 0 would mean that the proposed approach, which is also PPG-based, performs worse than Empatica's algorithm, whereas an error of 0 would just indicate that the proposed approach is as good as Empatica's algorithm. How can this approach be used to establish the superiority of the proposed method?
- It is unclear which set of parameters from Table IV were used in each of the combinations Comb1-Comb13.
- There is some inconsistency in the acceleration cap mentioned in section IV.C. A value of 0.2 is stated. However, looking at Table IV, it seems that the values for acceleration cap start from 0.1 with a step size of 0.85. How is it possible to have a value of 0.2? The end value of 0.05 in Table IV also seems erroneous and inconsistent.
- It is difficult to understand which results from Table VII and Table IX refer to TDCO and which to COSTF.
- Minor comment: It would be helpful to add some arrows in the Table column labels indicating whether a lower or a higher value is better.

---

### Official Review · Reviewer_TSLa · 2025-07-18
**Hybrid Adaptive Filters on Smartwatch PPG**

**Confidence:** 3
**Clarity Of Writing:** good
**Clinical Significance:** good
**Methodological Novelty:** great
**Overall Rating:** 5
**Final Rating:** 5

**Experiments And Results:**

fair

**Questions For The Authors:**

1.	Could you clarify the rationale for choosing only male participants in a narrow age range (26–29)?
This restriction may limit generalizability. Understanding whether physiological or logistical factors motivated this decision would help contextualize the findings. Expanding demographic diversity in future datasets could strengthen the work.
2.	How do you anticipate the model performing in the presence of irregular or pathological breathing patterns (e.g., apnea, dyspnea)?
A response to this would clarify the algorithm’s robustness to clinical use cases. This could significantly affect the perceived applicability of the system in real-world healthcare.
3.	Have you evaluated computational efficiency and memory usage of your adaptive filtering pipeline on embedded processors?
This would influence deployment feasibility on smartwatches. A brief profiling of latency or energy consumption might be sufficient and could raise the impact of your study.
4.	Can you provide insight into why TDCO and COSTF fail to benefit from your adaptive filtering scheme?
Understanding this interaction may highlight important design considerations when integrating motion artifact removal with other RR estimation strategies. A more detailed failure analysis might also improve the reproducibility of your results.

**Strengths:**

Major strenghts of the paper are:
•	High Practical Relevance: The study addresses an important challenge—motion-robust RR monitoring from smartwatches—enabling more reliable use in real-life health applications.
•	Technical Innovation: The hybrid use of EMD-derived synthetic reference signals for guiding LMS filtering is original and addresses a key limitation of standard adaptive filters.
•	Thorough Parameter Sweep: The exploration of 13 parameter configurations with performance visualization ensures reproducibility and robustness of conclusions.
•	Custom Evaluation Metric (FoM): The introduction of a composite FoM score that balances accuracy, precision, and estimation coverage provides a more nuanced comparison across methods.

**Summary Of The Paper:**

This paper proposes a novel signal processing framework for accurate estimation of respiratory rate (RR) using smartwatch-derived photoplethysmography (PPG) signals, specifically under motion-contaminated conditions. The authors develop and evaluate a hybrid approach that combines Least Mean Square (LMS) adaptive filtering with Empirical Mode Decomposition (EMD) to enhance the separation of motion artifacts from respiratory signals. Two versions of the algorithm are presented: a baseline frequency-domain approach and an enhanced adaptive version with motion artifact removal.
A small but controlled dataset was collected using the Empatica E4 smartwatch, including different breathing rates and motion scenarios. The authors evaluate their algorithm across 13 parameter combinations involving acceleration signal transformations, window lengths, step sizes, and motion cap thresholds. Performance is quantified using a custom Figure of Merit (FoM) metric that combines mean absolute error, standard deviation, and completeness of estimation.
The proposed adaptive method demonstrated a 71% reduction in mean error and a 56% reduction in standard deviation compared to the best non-adaptive alternatives. It also outperformed several re-implemented baseline RR estimation algorithms both with and without motion artifact removal.

**Weaknesses:**

However, several weaknesses arise from the study:
•	Small Dataset with Limited Demographic Diversity: The study relies on a dataset from only four healthy male participants (ages 26–29), which limits the generalizability to other populations, especially those with respiratory conditions.
•	Lack of External Validation: No testing is done on public or independent datasets. While real-world motion scenarios are simulated, external validation would better demonstrate scalability.
•	Minimal Clinical Framing: The clinical utility of high-resolution RR estimation (e.g., for detecting apnea, COPD monitoring, or stress assessment) is not sufficiently discussed or framed.
•	Black Box around Feature Fusion (SFU): Although SFU is used to combine AM, BW, and FM features, the decision rules and failure modes could be better explained or visualized.
•	Computational Cost: While acknowledged, the runtime implications of using EMD and adaptive filtering on smartwatch hardware are not analyzed. Real-time feasibility remains unclear.

---

### Official Review · Reviewer_PLtU · 2025-07-19
**The authors contribute a new method for RR estimation that outperforms existing algorithms.**

**Confidence:** 4
**Clarity Of Writing:** great
**Clinical Significance:** good
**Methodological Novelty:** great
**Overall Rating:** 8

**Experiments And Results:**

great

**Questions For The Authors:**

Have the authors considered running their experiments with window sizes of ranging from 5 to 29 seconds, incremented by 2?

**Strengths:**

The authors submit a well-written report that is easy to navigate and address the issue of motion artifacts in efforts to accurately estimate RR. The proposed algorithm demonstrates marked performance in both accuracy and reliability. Moreover, the method has promise as a practical solution for continuous RR monitoring in real-world free-living conditions.

**Summary Of The Paper:**

The authors propose an adaptive algorithm for respiratory rate (RR) extraction from photoplethysmogram (PPG) signals that were recorded using a smartwatch. The proposed algorithm combines least mean squares (LMS) filtering with a reference signal from empirical mode decomposition (EMD), including spectral analysis, for RR estimation.

**Weaknesses:**

I recommend that the authors assess agreement between algorithms using the limits of agreement (LOA) method (e.g., P. Charlton et al., 2026; Physiol Meas). This may supplement their figure of merit (FoM) method; importantly, it would make it easier to compare findings in this study with findings from the literature.

The authors may consider determining whether the respiratory modulations in their PPG signals are sufficiently strong for accurate RR estimation (D. A. Birrenkott et al., 2017; IEEE Trans Biomed Eng).

How do the authors expect the proposed algorithm would perform when applied to longitudinal data; which is the type of data typically collected using wearable smartwatches.

And how would that impact the efficiency of computation for both the base and adaptive versions?

The authors may consider discussing the small sample of N=41 as a limitation.

Currently it is unclear what parameters were used for spectral analysis; please add this detail.

It is also unclear how the SVD was used in this work to generate a reference signal; please clarify.